

# TOC intercomparison of Brewer, Dobson and BTS Solar at Hohenpeißenberg and Davos 2019/2020

Ralf Zuber[1], Ulf Köhler[2], Luca Egli[3], Mario Ribnitzky[1]. Wolfgang Steinbrecht[2], Julian Gröbner[3]

[1]Gigahertz-Optik GmbH, Türkenfeld/Munich, 82299, Germany
[2]Deutscher Wetterdienst (DWD), Hohenpeißenberg, 82383, Germany
[3]Physikalisch-Meteorologisches Observatorium Davos (PMOD/WRC), 7260 Davos Dorf, Switzerland

*Correspondence to*: R. Zuber (r.zuber@gigahertz-optik.de), L. Egli (luca.egli@pmodwrc.ch), U. Köhler (ulf.koehler@dwd.de)

**Abstract.** In the 2019/2020 measurement campaign at Hohenpeißenberg (Germany) and Davos (Switzerland) we compared the well-established Dobson and Brewer spectrometers (single and double monochromator Brewer) with newer BTS array
spectroradiometer based systems in terms of total ozone column (TOC) determination. The aim of this study is to validate the BTS performance in a longer-term TOC analysis over more than one year with seasonal and weather influences. Two different BTS setups have been used. A fibre coupled entrance optic version by PMOD/WRC called Koherent and a diffusor optic which proved to be simpler in terms of calibration from Gigahertz-Optik GmbH called BTS Solar. The array-spectrometer based BTS systems have been traceable calibrated to National Metrology Institutes (NMI) and the used TOC retrieval algorithms are
based on spectral measurements in the range of 305 nm and 350 nm instead of single wavelength measurements as for Brewer or Dobson. The two BTS based systems, however, used fundamentally different retrieval algorithms for the TOC assessment, whereby the retrieval of the BTS solar turned out to achieve significantly smaller seasonal drifts. The intercomparison showed a deviation of the BTS Solar to Brewers of <0.1% with an expanded standard deviation of <1.5% within the whole measurement campaign. Koherent showed a deviation of 1.7% with an expanded standard deviation of 2.7% mostly given by
a significant seasonal drift. Resulting, the BTS Solar performance is comparable to Brewers at the comparison in Hohenpeißenberg. The slant path slope is in-between double monochromator and single monochromator Brewer. Koherent shows a strong seasonal variation in Davos due to the sensitivity of its ozone retrieval algorithm to stratospheric temperature similar to the Dobson results.

## 1 Introduction

Atmospheric ozone has been defined an essential climate variable in the global climate observing system (GCOS) of the WMO. Careful long-term monitoring of the global ozone layer is still crucial in verifying the successful implementation of the Montreal Protocol and its amendments (MPA) on the protection of the ozone layer, with the eventual recovery of the ozone layer to pre-1970's levels.

McKenzie et al. (2019) showed for instance that no global, statistically significant recovery of the ozone layer took place yet,
but no further decline either. This alone suggests that longer periods of observation are necessary. In addition Montzka et al. (2018) stated that despite the ban, CFC-11 has been released in measurable quantities in China. This means that not only the control of the expected recovery of the ozone layer, but also the monitoring of the protocol for the CFC ban are important tasks (Bais et al., 2019).

On the other hand, the effects of the global climate change to the ozone layer are yet not completely understood (Bais et al.,
2019;Bais et al., 2018;Bais et al., 2015;Seckmeyer et al., 2018) which is another argument why further observations will be necessary.

The monitoring of the ozone layer started already in 1926, when the at that time "modern" Dobson Spectrophotometer was developed and constructed by Prof. G.M.B. Dobson at the Oxford University (Dobson, 1931;Dobson, 1968). A first small network with six stations (Abisko, Arosa, Lerwick, Lindenberg, Oxford, Valentia) was established in 1926 (Köhler and Claude,



2006), to measure the total ozone column (TOC). A remarkable extension of the global network took place since 1957/1958 (International Geophysical Year). In the following decades, up to 130 instruments were produced and employed. The Meteorological Observatory Hohenpeißenberg (MOHp) of the German National Weather Service (DWD) started its observations of the ozone layer with ozone sonde (vertical profile of the ozone up 35 km) and with Dobson No. 104 (TOC) in 1967/68. Approximately 50 stations with Dobson instruments are still operational today and submitting data to the World

Ozone and UV Data Centre (WOUDC) in Toronto (Canada). Thus, this type of spectrophotometer is still a backbone in the global ozone monitoring network, although new, modern instruments like the Brewer Spectrophotometer have been established in this network since the 1980s.

Today Dobson and Brewer spectrophotometers are the main instruments used to monitor the ozone layer, and have been in use since the 1920's and 1980s respectively. Dobson relevant papers are: Dobson (1931), Dobson (1957b), Dobson (1957a),

Dobson (1968), Komhyr (1980) and Evans (2008). Publications about the function of Brewer spectrometers are: Kerr et al. (1981), Kerr et al. (1985), Savastiouk (2006). Many publications about long term intercomparisons between both spectrophotometer types have been published in the past three decades: Kerr et al. (1988), Kö◌̈hler and Attmannspacher (1986), Köhler et al. (1989), Staehelin et al. (2003) with a detailed description of function and ozone retrieval algorithms of Dobson and Brewer, Scarnato et al. (2010). An important publication on the compatibility of Brewer measurements in Arosa

and Davos have been published by Stübi et al. (2017). The paper of Redondas et al. (2014) describes the effects of the planned introduction of new ozone absorption cross sections (Gorshelev et al., 2013;Serdyuchenko et al., 2013) on the Brewer and Dobson ozone retrievals. The comparison of Dobsons and Brewers with different cross-sections is newly performed at PMOD/WRC (Gröbner J., 2021). Both systems are still in use, even though Dobson spectrophotometers are no longer being manufactured. In order to further maintain a dense network of TOC observations worldwide, new automatic, reliable, simple

and cost-effective systems have been developed.

State-of-the-art array spectrophotometers measuring spectral UV radiation are potential candidates for such new instruments (Egli et al., 2016). Due to strong ozone absorption between 300 nm and 350 nm, TOC can be retrieved from spectral direct sun measurements in the UV band (Huber et al. 1995). The same spectral range is used for TOC retrieval from Brewers and Dobsons (Kerr et al., 1988). However, the advantage of array spectroradiometers is that a continuous spectral range is measured

instead of only discrete wavelength values as for the Brewer (5 wavelengths) or the Dobsons (2 and 2x2 wavelengths, respectively). It is expected that this additional spectral information increases the reliability of TOC retrieval. The general specifications for spectral global solar UV measurements in the TOC relevant band are stated in several guideline-publications (e.g.: (Seckmeyer *et al.*, 2010; Seckmeyer *et al.*, 2001). The performance of global UV measurements of different commercially available array-spectroradiometer have been tested within an intercomparison in Egli *et al.*, 2016. The results of

the global UV intercomparison in Egli et al. 2016 showed that the tested instruments suffered significantly from straylight. In particular at high solar zenith angles, the UV radiation at short wavelengths between 300 nm and around 315 nm are significantly biased, while the spectral measurements at longer wavelength are well reproduced. Just at low SZA some well characterized and calibrated devices provided acceptable deviation in the range of 5% to the reference.

First instruments using the full spectrum of array spectroradiometer for TOC retrieval are Pandora (Herman et al., 2015) and

Phaeton (Gkertsi et al., 2018). The Pandora system retrieves TOC by spectral fitting of the attenuated spectrum in the wavelength band between 305 nm and 330 nm. Herman et al. (2017) reported an averaged difference of 2.1 ± 3.2% for TOC with a relative drift of 0.2% per year of the Pandora system #34 compared to the NOAA Dobson #061 in Boulder Colorado based on a three years intercomparison. For the comparison, both instruments have been corrected for the impact of stratospheric temperature using their standard ozone absorption cross section and for the effect of straylight a correction has

been implemented for the Pandora system.

The Phaeton instrument is a DOAS/MAXDOAS system using the wavelength band between 315 nm and 337 nm. The instrument was characterized for straylight with a tuneable laser and a straylight correction was applied. The two-years





comparison with a single monochromator Brewer spectrophotometer revealed an average bias of 1.85 ± 1.86% for TOC using the Paur and Bass (1985) cross-section at constant temperature of 228K. When including the daily variability of the

stratospheric temperature the differences are reduced to 0.94 ± 1.26%.

The BTS array spectroradiometer has been released after the quality assessment of aforementioned global UV array spectroradiometers. The BTS has been compared separately to well established double monochromator-based systems in terms of solar global spectral irradiance (Zuber et al., 2018a). The BTS reduces straylight by filtering optical radiation from longer wavelength with different bandpass filters inside the array-spectroradiometer. Due to this straylight suppression, the UV index

measurements derived from the BTS spectra were within ±1% for solar zenith angles (SZA) smaller than 70° and ±3% for SZA between 70° and 85° in reference to an NDACC device (Zuber et al., 2018a). The comparison of the BTS solar UV spectra with the reference showed some biases of up to 5% for short wavelengths at wavelengths shorter than 310 nm for SZA larger than 70° (Zuber et al., 2018a). These biases may affect the TOC retrieval using the full spectrum, including short wavelengths. Addressing this question, Zuber *et al.*, 2018b noted a deviation of TOC determination of less than 1.5% (for SZA

smaller than 65°) to other instruments in most situations and not exceeding 3% from established TOC measurement systems such as Dobson or Brewer at the Izaña Atmospheric Observatory

in Tenerife. At higher SZA the deviation slightly increased for the TOC determination. This may be an effect of the remaining straylight in the wavelength band between 305 nm and 310 nm, which was used during this intercomparison. Potential effects on TOC caused by the remaining straylight will be further discussed based on the results of his study.

In this publication we show the longer-term performance of BTS based systems concerning TOC determination in terms of different solar zenith angles, different TOC level, seasonal and weather influences in harsh environment at Hohenpeißenberg (Germany) and Davos (Switzerland) for a period of more than one year. Furthermore, two distinct different retrieval algorithms from the BTS Solar and the Koherent system are compared and discussed based on the results of this long term intercomparison.

**2 Measurement campaign, instruments and retrieval**

The intercomparison took place in the 2019/2020 measurement campaign at the Deutscher Wetterdienst DWD Hohenpeißenberg, Germany (altitude: 953 m; coordinates: 47.80090 N, 11.010891 E) and Physikalisch-Meteorologisches Observatorium Davos (PMOD/WRC), Switzerland (altitude: 1580 m; coordinates: 46.81 N, 9.83 E).

Both stations are part of the Global Ozone Monitoring Network of WMO/GAW (World Meteorological Organisation / Global

Atmosphere Watch) and the European Brewer Network (http://www.eubrewnet.org/). The DWD Hohenpeißenberg has been the WMO Regional Dobson Calibration Centre for Europe (RDCC-E) since 1999 and belongs as Global Station to the GAW-network since 1995 (Köhler, 2002). The PMOD/WRC is the world calibration centre for meteorological radiation measuring instruments. At Hohenpeißenberg the BTS Solar is compared to the instruments Dobson 104, Brewer 010 (single monochromator) and Brewer 226 (double monochromator). At Davos the BTS based Koherent is compared to the Brewer 163.


**2.1 Dobson/Brewers**

The operational standard instruments, used at the Hohenpeißenberg Observatory to measure the thickness of the ozone layer are one Dobson spectrophotometer and two Brewer spectrophotometers, both types of spectrophotometer use a similar physical principle. They measure the incoming solar radiation in various spectral lines in the short UV-B (280 – 315) nm and the longer

UV-A range (315 – 400) nm. Whereas the short wavelengths are affected by ozone absorption, the longer wavelengths longer than 340 nm are unaffected by ozone absorption.

The Dobson spectrophotometer measures the relative intensity of the solar radiation at two wavelengths, using a photomultiplier tube (PMT) and a so-called optical wedge. This optical wedge is slid by a wedge dial into the beam of the



strong, longer wavelength, until both signals become equal. This means that the difference of the currents alternating produced in the PMT become zero. The balance position of the dial corresponds to a well-known attenuation derived from a special two-lamp wedge calibration. This measured attenuation simulates the absorption of the short UV-radiation by the ozone layer and thus the optical wedge can be considered as a kind of virtual ozone layer inside the Dobson. This relationship of the radiation intensities in the long and short wavelength is a direct measure of the thickness of the ozone layer. Using the physics describing the ozone absorption in the UV mathematics based on the beer-lambert extinction law can be applied to calculate the thickness of the ozone layer between the instrument and the top of the atmosphere (Komhyr, 1980). The use of two wavelength pairs (called AD) with the double ratio technique significantly reduces atmospheric effect like the Mie- and Rayleigh absorption, caused by atmospheric variations such as cirrus clouds or aerosol absorption and scattering. In contrast to the single ratio of one wavelength pair, the double ratio technique using two wavelength pairs is mostly insensitive to these additional effects and is therefore currently considered the method of choice for the total column ozone retrieval (Kerr, 2002;Staehelin et al., 2003). Thus the impact of the mie-scattering can be minimized (Vogler et al., 2007). The last official calibration services of the Dobson No. 104 with the Regional Standard Dobson No. 064 of the Hohenpeißenberg RDCC-E were performed in 2018 and 2019 and confirmed its very good calibration level.

The measurement principle of the Brewer spectrophotometer is similar, as it uses the spectral lines of the solar radiation in the same UV-range as the Dobson. As the Dobsons, the TOC retrieval with Brewer is obtained with the double ratio technique of the wavelength pairs using a slightly different weighting for the individual channels (Kerr et al., 1985;Kerr et al., 1981). The essential difference is that this spectrophotometer measures the photon counts in each of the used five wavelengths (four for $O_3$, one for $SO_2$) directly with a photomultiplier tube. The same procedure of the double ratio technique as with the Dobson is then applied to derive the TOC. One advantage of the Brewer type is its automatic operation using a PC with a special control and TOC-calculation software, which is in contrast to normal Dobsons, that are manually operated (except the Swiss automatic Dobsons, Stübi et al. (2017) ). However, both methods of measurements, manual or automatic, need several minutes (Brewer e.g. allow a single measurement within 30 seconds, however typical 3 minutes are needed in order to detect cloud influences) to obtain a reliable value of the TOC. This temporal constraint limits the number of possible observations on days with unsettled weather conditions and fast-moving clouds. The BTS based devices allow measurements within 8 to 45 seconds (depending on SZA and BTS setup), however usually an averaging of 1 min to 5 min is applied in order to improve the signal to noise ratio.

The Hohenpeißenberg Brewers No. 010 (running since 1983) and 226 (running since 2014) have undergone a calibration service in July 2020, done by IOS using the traveling standard Brewer No. 017 (see Figure 2). These kind of calibration services have been performed every year in the past 20 years. The stability of the calibration levels of both Brewers (differences to the reference <1%) have been confirmed almost every year, so in 2020 too.

### 2.2 BTS Solar

BTS Solar is based on a BTS unit with directly mounted diffusor entrance optic and a tube for FOV limitation for irradiance measurements (no optical fibre). The device is described and characterised in detail in Zuber et al. (2018b). For the TOC determination in principle a full least square algorithm in the wavelength range from 305 nm to 350 nm according to Huber et al. (1995) could be used to take advantage of the entire full spectrum, as is used by the Koherent measurement system discussed in section 2.3. However, for the BTS Solar we use a different TOC retrieval algorithm in order to validate its long-term/seasonal performance. The algorithm is also described in more detail in the Zuber et al. (2018b). In summary, the technique is based on the comparison from two selected wavelength bands (instead of the entire spectrum as for Koherent) of the measured data with a lookup table (LUT) pre-calculated by the libRadtran software package for radiative transfer calculations (Emde et al., 2016).





For the intercomparison within this paper two adaptions to previous campaign retrieval algorithm (Zuber et al., 2018b) have been applied.

First, the two wavelength-bands which are considered for the TOC retrieval are adapted to 307 nm to 311 nm (high ozone absorption) and 319 nm to 324 nm (low ozone absorption, see Figure 1), which showed to be the most robust configuration

for the BTS device. Reason for this choice is that both wavelength bands are within one optical filter measurement of the BTS and the measurement time could be further decreased in principle to this single filter measurement. Second, an air pressure correction for this retrieval algorithm has been introduced in order to consider the larger differences that can be expected within a period of the measurement campaign. A correction model has been used since the calculation of a different LUT for every air pressure would be possible but too calculation-time extensive and the correction within the whole expected air pressure

dynamic is <0.8 DU (for an air pressure range of 870 hPa to 930 hPa as observed at Hohenpeißenberg, a correction of maximum 2.5 DU results). For this purpose, five different LUTs with different pressures where calculated and a best fit model for the pressure correction was determined. This correction has been applied after the TOC retrieval based on a standard pressure with the actual present air pressure during the measurement campaign. The air pressure data has been supplied by the DWD.

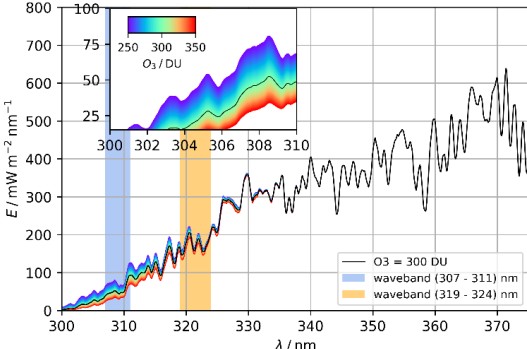

**Figure 1: Direct irradiance spectra of the lookup table modelled with libRadtran. The two wavebands for the TOC retrieval are marked with blue (high O3 absorption) and orange (low O3 absorption)**

In general, for the calculation of the LUT, the following values for the input parameters have been chosen for Hohenpeißenberg: air pressure as stated above, an altitude of 953 m, an atmospheric profile typical for subarctic (due to the mountain region) summer and winter (transition in April and October) given by Anderson et al. (1986), the ozone cross section

of Paur and Bass (1985) (the IUP cross-sections of Serdyuchenko et al. (2013) are in the meantime also available for libRadtran). A sample check of one day (24.02.2020) results in a difference of 0.4 DU when using Bass and Paur or IUP. Finally, the high resolution QASUMEFTS extra-terrestrial spectrum of Gröbner et al. (2017) is used for the libradtran modelling. For aerosol, albedo (which has no significant influence to direct solar irradiance) and SO$_2$ standard values of libRadtran have been used. This methodology was applied with excellent results to measurements at Izaña Atmospheric Observatory in Tenerife (Zuber

et al., 2018b) using slightly different wavelength for the ozone retrievals. As well at the intercomparison in Huelva with comparable results which have not been scientifically published yet. The results of this study with the adopted wavelength show that a retrieval based on a comparison of the LUT to the measurement provided accurate results for many cases. At this point we want to emphasize that this retrieval method based on LUT does not require any further time dependent input parameters apart from the current air pressure. However, a more precise and extensive modelling also for each measurement

individually, would be possible and maybe also needed for some measurement sites with different and fast changing atmosphere. However, in the libradtran software package further atmospheric profiles are available and can be applied for other sites (midlatitude summer and winter, subarctic summer and winter, tropical and US standard atmosphere) (Emde et al., 2016). During this campaign, the mentioned pre-defined standard parameters were used for the LUT and were not changed within the whole intercomparison campaign except for the seasonal atmosphere model (summer/winter), which was changed



in April and October. At the date of the season change no significant change in TOC retrieval was recognizable. No additional correction of stratospheric ozone temperature has been applied so far.

The BTS Solar has been calibrated for this campaign with a well-known spectral irradiance calibration setup in 70 cm distance at the Gigahertz-Optik ISO 17025 certified calibration laboratory (D-K-15047-01-00, D-PL-15047-17025-2020). This calibration showed to be robust and small measurement uncertainties of 2.5% (expanded measurement uncertainty k=2) in

spectral irradiance can be achieved (Zuber et al., 2018b;Vaskuri et al., 2018). After 6 month a check of the calibration has been performed, which showed no significant change (differences below the calibration uncertainty of the standard lamp). Resulting the calibration has not been changed within the whole measurement campaign.

A measurement interval of 2 minutes has been chosen. During the whole measurement campaign at Hohenpeißenberg, only

16 days were affected from power breakdowns due to construction work at the site and an IP server issue/software conflict in the beginning of the campaign. These are about 3% downtime during the whole campaign. For the performance evaluation all measurement data of the whole intercomparison campaign were considered. Only measurements with a standard deviation error of 10 DU within five consecutive measurements have been removed since such a large change in TOC within such a short time interval can only be expected due to instrument malfunction, or cloud movement or very high SZA.


### 2.3 Koherent

Koherent is based on a fibre coupled BTS-2048-UV-S-F array spectroradiometer. Contrary to the instrument at Hohenpeißenberg, the BTS-2048-UV-S-F is connected to a lens-based imaging telescope, with a field of view of about +/-0.6°. The telescope is mounted on a sun tracker on the measurement platform at PMOD/WRC as presented in Egli and Gröbner

(2018). The array spectroradiometer is embedded in a temperature stabilized weather-proofed box keeping the instrument at a constant temperature of $(22 \pm 1)$ °C through all seasons of the year.

The raw data of the BTS is acquired with a python software-routine on an embedded computer inside the box, providing data with almost no technical failure during the entire campaign. The readings are taken with an interval of 1 min for each measurement. Usually, the device needs around 45 seconds for one measurement. However, at high SZA in the early morning

or late evening with low direct sun irradiance intensity, around 2 minutes are needed due to increased integration time. The data was aggregated to a 5 minutes data-point, by averaging the individual data. If the standard deviation of the individual measurements within 5 minutes exceeded 10 DU, the data was considered as invalid. These small-time-scale variations are caused by instable atmosphere such as cirrus clouds or noise from the detector at low sun elevations. The criterion of 10 DU was chosen to remove theses outliers.

The postprocessing was performed off-line with the following steps: a) converting the raw-counts of the readings to irradiance using the laboratory calibration, b) wavelength shift correction using the MatShic software developed at PMOD/WRC (Egli, 2014) and c) retrieving TOC with a least squares minimization algorithm according to Huber et al. (1995) and Vaskuri et al. (2018).

The aforementioned least squares algorithm is using a full spectral fit in the wavelength range from 305 nm to 350 nm and an

atmospheric model based on the Beer-Lambert law:

$$I_\lambda = I_0 \, exp[-\tau_\lambda m]$$

$I_\lambda$ represents the measured solar irradiance from Koherent at the specific wavelength $\lambda$ and $I_0$ is the extraterrestrial spectrum at the top of the atmosphere. For the retrieval, the QASUMEFTS extra-terrestrial spectrum is applied (Gröbner et al. 2017). The airmass $m$ is important for the attenuation of ozone through the atmosphere, which depends on the time of the day and

thus on the solar zenith angle during a day and the seasons. For the calculation of the airmass a constant height of the ozone layer at 22 km is used. The absorption is summarized by the term $\tau_\lambda$, indicating the absorption by ozone, aerosols and Rayleigh scattering. The absorption cross section by ozone is provided by University of Bremen, IUP (Serdyuchenko et al., 2013). For



the retrieval, the IUP cross-section at the stratospheric temperature of constant -45° is used for the standard US atmosphere afglus. The assumptions of -45°C is similar to the standard Brewer procedure.

Finally, the aerosol absorption is parametrized with a linear parametrization and the Rayleigh scattering by the parametrization of Nicolet (1984) assuming a constant ground air pressure of 820 hPa (for the higher altitude station Davos).

The advantage of the minimal least square fit approach is that the attenuation by aerosol optical depth and the effect of Rayleigh scattering are retrieved for each measured data during the ozone retrieval and are not input parameters for the retrieval. The stratospheric temperature and the height of the stratospheric layer are currently not retrieved by the algorithm as well as $SO_2$

is not included. In summary, the retrieval algorithm used for this comparison, displays a first simple approach to test the performance of the above standard settings of the retrieval method.

The system was calibrated for absolute irradiance at the PMOD/WRC laboratory facilities on 14[th] and 19[th] of November 2019 using four 1000W FEL lamps on a linear stage at distances of 2.40 m and 1.30 m. Due to the low light-throughput of the fibre-coupled telescope setup and the sensitivity of the alignment on small angular deviations on the linear stage, the responsivity

varied resulting in TOC differences of 2.2% (*k=1*). An averaged responsivity of the system, which preliminary appeared to be the best calibration, was used for the calibration of the raw data and resulting TOC from Koherent presented here.

In analogy to the comparison at Hohenpeißenberg, Koherent is compared with the PMOD/WRC double monochromator Brewer 163. The Brewer is annually calibrated for TOC retrieval by the Regional Brewer Calibration Center Europe (RBCC-E) with an uncertainty of around 1%, while the stability of the instrument is operationally monitored on a daily schedule. Due

to this quality and stability assurance of the Brewer 163 TOC data, the Brewer 163 serves as a reference for the comparison with Koherent in Davos. Brewer TOC readings are taken on an irregular schedule, since the Brewer also measures operationally spectral global UV irradiance.

## 3 Results of the Intercomparison

### 3.1 BTS Solar at Hohenpeißenberg


A special day of the intercomparison at Hohenpeißenberg (July 28, 2020) is presented in Figure 2. During this day the RDCC-E reference Dobson No. 064, the Indian Dobson No. 036 and the IOS Brewer No. 017 for service measurements purposes took additionally part. The measurement of seven instruments (BTS Solar, Dobson No. 036, 064 and 104, Brewer No. 010, 017 and 226) showed a good agreement with differences between all instruments mostly less than ±1% (see figure 2) on this special

day. Only at low sun elevations, in the morning the BTS solar and the single monochromator Brewer 010 show a bias of about 5 DU to the other instruments.

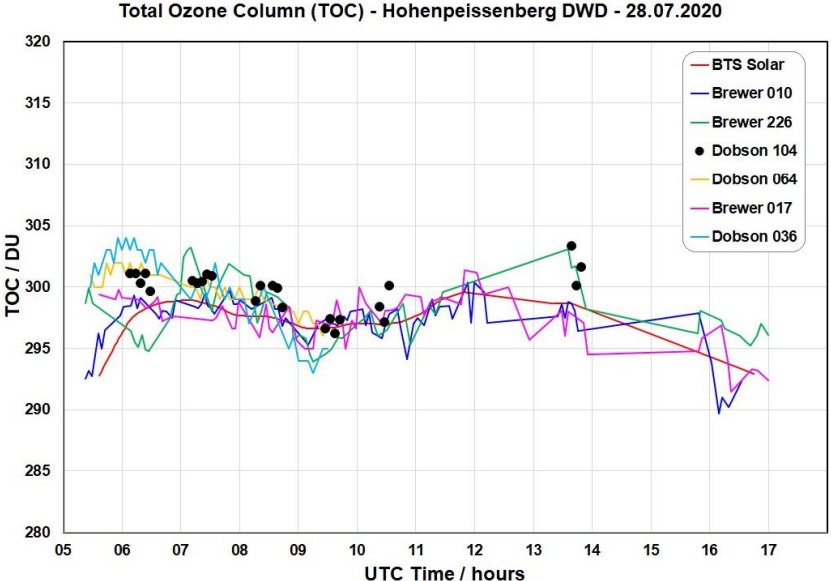

**Figure 2 TOC measurements of the IOS Brewer calibration service on the 28.07.2020 at Hohenpeißenberg**

In the following, Figure 3 presents two exemplarily daily courses of TOC, which show a strong diurnal dynamic in TOC and

the high temporal resolution of the devices compared to the Dobson. These data show that the BTS Solar and both Brewers allow about the same TOC determination in time (SZA). The Dobson shows the aforementioned offset in winter times and delivers only a few data points for each day due to its manual operation.

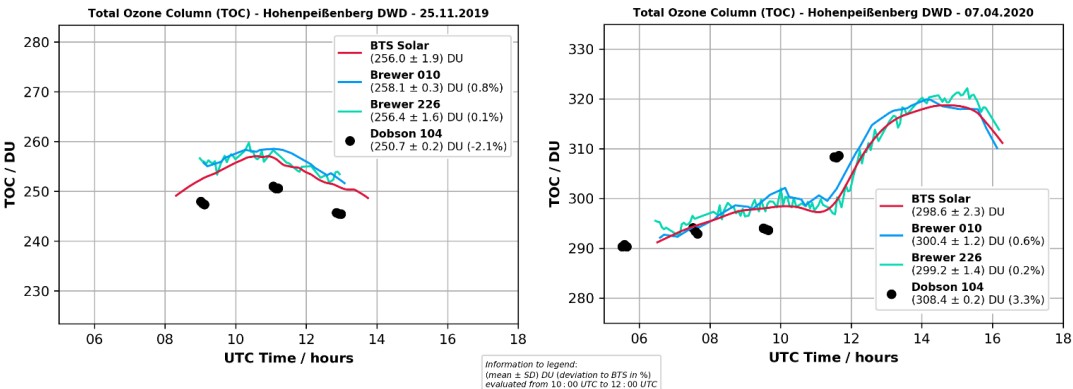

**Figure 3 Two exemplarily days of the intercomparison which show a strong dynamic in TOC within the day.**

Since different devices are compared to the BTS Solar and all of them exhibit their own measurement uncertainty (e.g. 3.7 DU for the BTS, see Vaskuri et al. (2018)) we compared all data in Figure 4 relative to the BTS Solar.

Figure 4 shows that the histogram of all data is comparable between the Brewer 010 with a mean deviation of 0.04% (percental deviation between the devices (e.g. (Brewer010-BTS Solar)/BTS Solar·100) and an expanded standard deviation ($k=2$) of

1.29% and Brewer 226 with a mean deviation of 0.06% and an expanded standard deviation ($k=2$) of 1.47%. The Dobson 104 shows a mean deviation of -0.84% and an expanded standard deviation ($k=2$) of 2.22%.




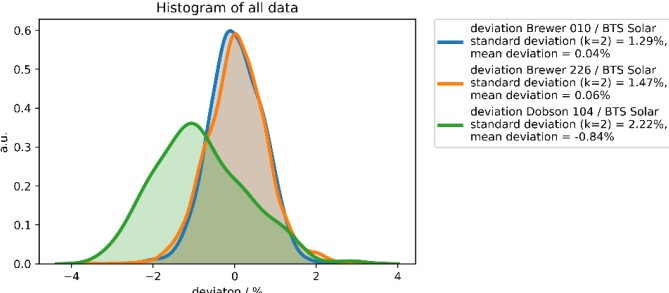

**Figure 4 Histogram of all Brewer 010, Brewer 226 and Dobson 104 compared to BTS Solar data from Hohenpeißenberg. The data shows a comparable performance between Brewer 010 (deviation of 1.29% to BTS Solar), Brewer 226 (deviation of 1.47% to BTS Solar) and BTS Solar and a worse performance of Dobson 104 (deviation of 2.22% to BTS Solar).**


In the following Figure 5 the seasonal course of all TOC data of the complete measurement campaign is illustrated. This explains the larger deviation of the Dobson 104 due to the seasonal dependency (Gröbner J., 2021;Kerr et al., 1988;Scarnato et al., 2010;Staehelin et al., 2003;Vanicek, 2006). The plot also shows the comparable trend of the Brewers and BTS Solar, since the least square fit is within ±1% over the whole measurement campaign.


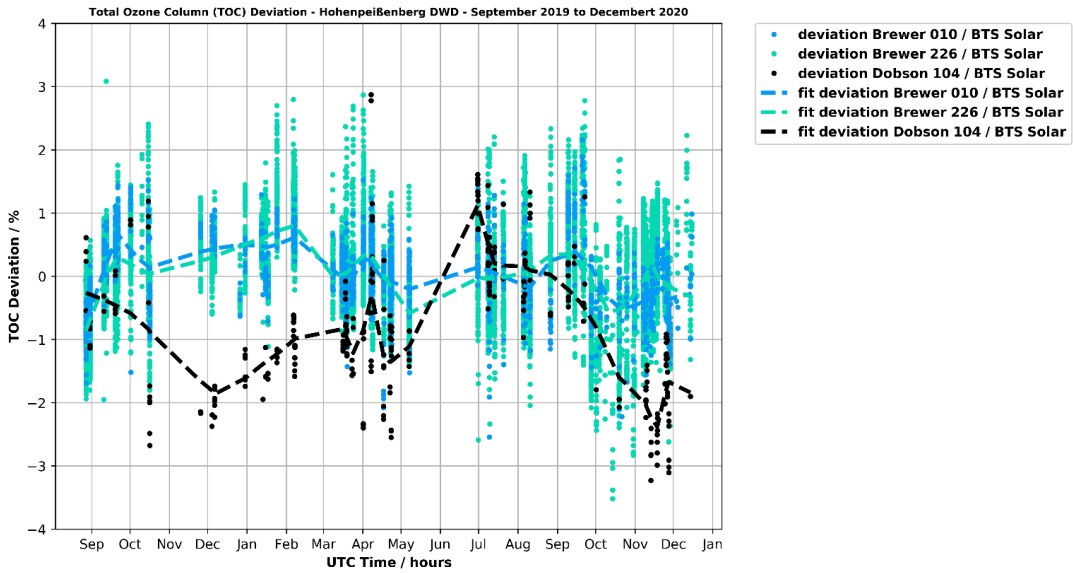

**Figure 5 Full measurement campaign TOC plot from Hohenpeißenberg. This plot shows the seasonal dependency of the Dobson 104 but it shows also the comparable temporal course of the Brewers and BTS Solar which is expressed by the least square fit deviation of less than ±1% over the whole campaign.**

Figure 6 shows the TOC slant path deviation of the Brewer 010 (single monochromator) and BTS Solar compared to the Brewer 226 (double monochromator). This figure is a measure of the TOC evaluation with respect to air mass factor (AMF). AMF is dependent on the solar zenith angle (SZA) due to the longer attenuation path through the atmosphere at high SZA and the deviations should ideally be independent on this factor. As illustrated the BTS Solar is less dependent on the slant path as the Brewer 010 (slope -0.00174 %/DU·AMF) but a small dependency (slope -0.00056 %/DU·AMF) is recognizable compared 305 to Brewer 226.





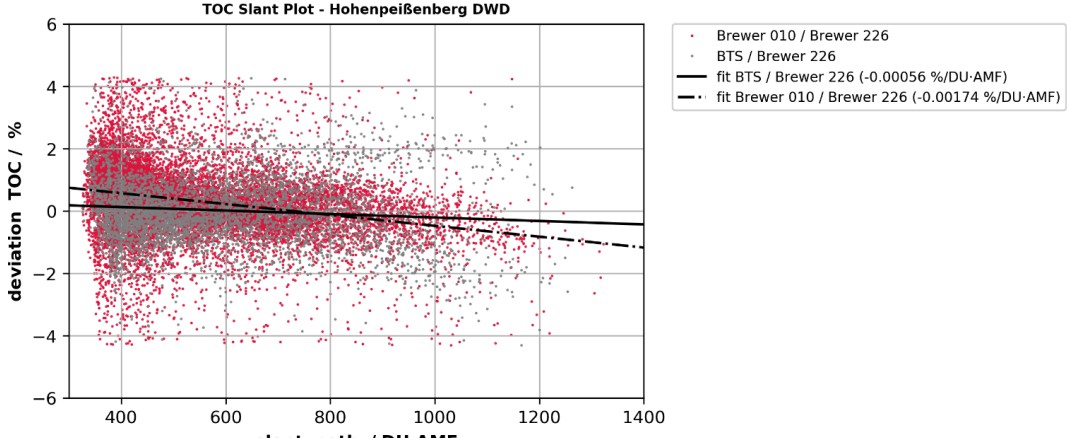

**Figure 6 The TOC Slant plot for BTS Solar of all data shows a smaller air mass factor (AMF) dependency than the single monochromator Brewer 010 for the BTS Solar. Linear fits are used.**

### 3.2 Koherent

Figure 7 presents exemplary daily courses of TOC from Koherent and Brewer 163 during the winter day of 2nd of February 2020 and the summer day of 27th July 2020, respectively.

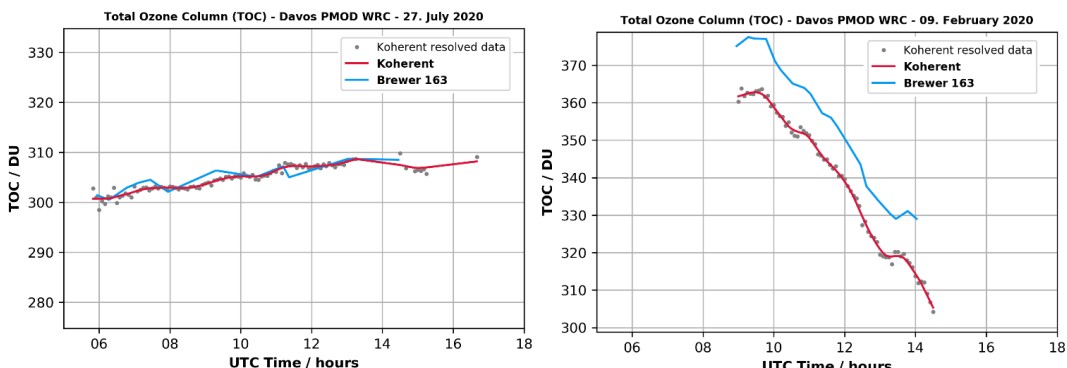

**Figure 7 Two exemplarily days of Koherent vs. Brewer 163 in winter and summertime. These days illustrate the high temporal**
**resolution of Koherent and show the seasonal dependency of the deviation.**

One can see that the daily variation is well reproduced by Koherent with a high temporal resolution and its temporal development is in line with the values of Brewer 163. However, there is an absolute bias of TOC of around -10 DU compared to Brewer 163 depending on the season. In wintertime Koherent shows a larger offset than in summertime, where the values are almost congruent.

Figure 8 shows the temporal course of the relative difference of TOC values between Koherent and Brewer 163 during the entire comparison period between 1st October 2019 and 30 December 2020 ((Koherent-Brewer163)/Brewer163). The least square fit in the figure reveals clearly the seasonal dependency of up to 4% of the differences through the seasons. The differences are about -3% in summertime and about +1% in wintertime in respect of the fit in figure 8.

Figure 9 presents the histogram of all synchronous data of the difference of Koherent and the Brewer 163. The histogram of
the percentual differences of TOC reveals a systematic overestimation of Koherent of a mean of 1.64% during the entire period, with an expanded standard deviation of about 2.7% (*k=2*). This variation reflects mainly the seasonal variation over the seasons





and less the measurement error of the system, which is assumed to be in the order as for the BTS Solar system (see section 3.1).

In order to compare the slant path dependency of the instrument, Figure 10 demonstrates the percental deviation of the differences as a function of the ozone slant path, defined as the multiplication of the AMF and TOC. Figure 10 a) shows a strong ozone slant path dependency with a slope of -0.0033%/TOC·AMF than is observed with Koherent. For further comparison, the slant path dependency is determined only for the summer month of July 2020 only (Figure 10 b)), with air masses between 1.1 and 3.4 and stratospheric temperatures ranging from 230.2 K to 231.1 K. The slope of the linear fit of -0.0015%/TOC·AMF indicates that the slant path dependency is significantly lower than for the entire period. Figure 10 b) also include a polynomial fit of degree=2 (quadratic), highlighting that for slant paths below 800 DU the slant path dependency shows almost no trend. For higher slant paths, however, the differences are increasing.

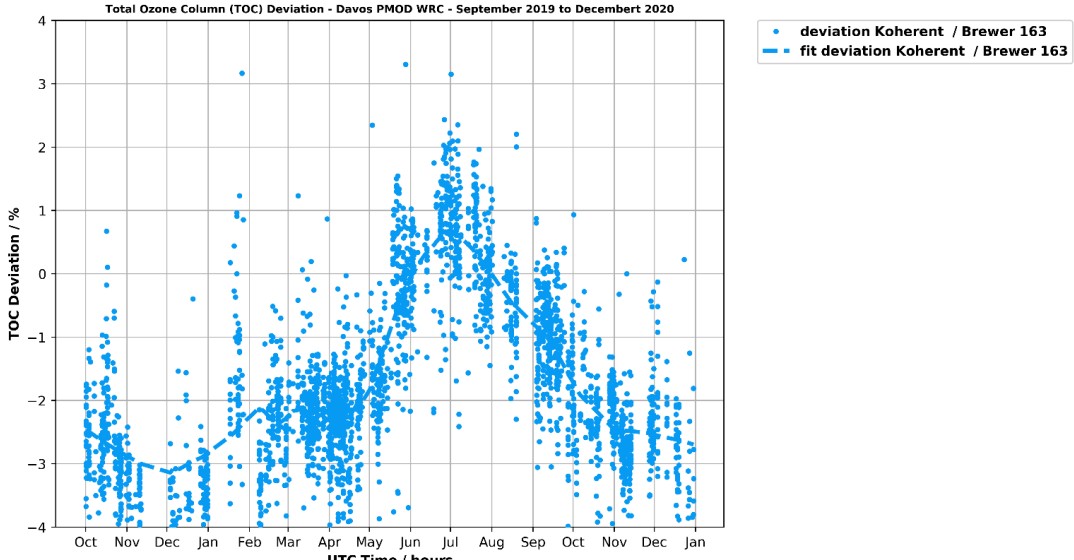

**Figure 8 Temporal course of the relative differences of Koherent and Brewer 163 during the entire period of comparison. The dashed line shows a least square fit of the data.**

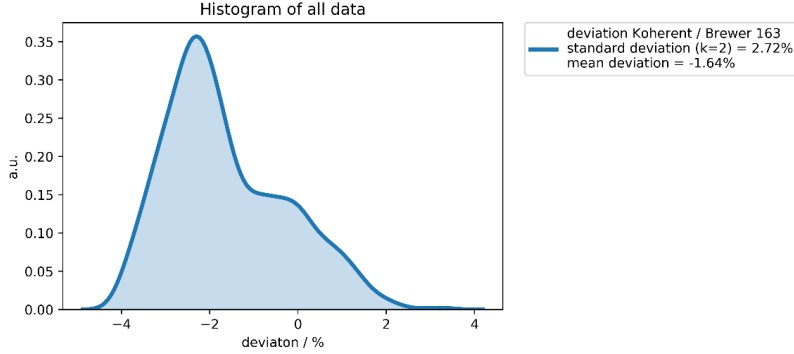

**Figure 9 Histogram of the relative differences of Koherent and Brewer 163. The mean difference between both instruments is 1.64% revealing a clear bias of TOC. The standard deviation (*k=2*) of 2.72%, reflects the seasonal variation of the deviation.**





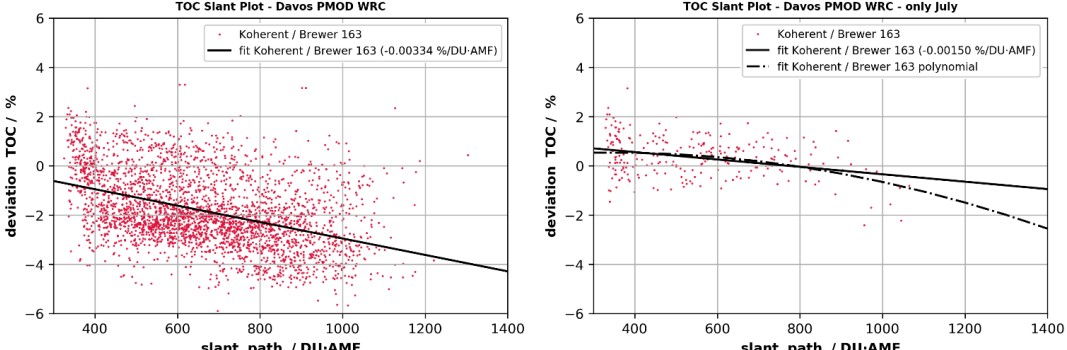

**Figure 10 a) and b). Slant path dependency of Koherent vs. Brewer 163 a) for the entire period of comparison and b) for the month of July 2020 only including a quadratic polynomial fit additional to the linear fits.**

**4 Discussion**

In this longer-term intercomparison of more than one year the array-spectroradiometer based BTS systems evidenced their performance published in Zuber et al. (2018b) and Egli and Gröbner (2018) for TOC determination as well during harsh environmental conditions like in Davos or Hohenpeißenberg. Both BTS setups, the fibre coupled entrance optic version by PMOD/WRC called Koherent and a diffusor optic based version by Gigahertz-Optik GmbH called BTS Solar where running the whole campaign without any significant technical issues.

The BTS Solar demonstrated to be robust and accurate in terms of stability since no change in responsivity was observed during the whole measurement campaign period (differences below the uncertainty of the calibration). The used TOC retrieval algorithm, based on a Look-Up-Table (LUT), demonstrated as well to be robust and accurate in terms of TOC determination over all seasons. Apart from the current air pressure (corrections of <0.8 DU over the full measurement campaign) no further parameters such as ozone temperature or other measured parameters are required as input of the algorithm for the evaluation at Hohenpeißenberg. Only standard parameters of the measurement location such as height and position are needed. Based on these parameters two (summer and winter) LUTs are calculated for a specific station with the libRadtran software package and later on just an air pressure correction is applied. This seems a remarkable, since already by this simple modelling accurate TOC results over the whole season have been achieved at this site.

The retrieval algorithm of BTS solar is principally a partial spectral fit of two wavelengths bands, which substantially differs from the four wavelengths retrieval from Brewers and double wavelengths pairs from Dobsons or the least square fit retrieval of Koherent. The partial spectral fit technique with the usage of a LUT demonstrated its performance so far in very different places such as Izaña Atmospheric Observatory in Tenerife (Zuber et al., 2018b), Huelva (not scientifically published yet) and now in Hohenpeißenberg. the algorithm allows worldwide usage (due to different available atmosphere models of libRadtran) but may need a finer parametrization at other locations worldwide or for special atmospheric profile situations which may not be tested yet. For the slant path dependency, a slope 0f -0.00056 %/DU·AMF was achieved, which is significantly lower than of single monochromator Brewer (0.00174 %/TOC·AMF) compared to double monochromator Brewers.

Contrary to BTS solar, the TOC retrieval from Koherent is using the full spectrum retrieval in the spectral range of 305 nm and 350 nm. The minimal least square algorithm as of Huber et al. (1995) and Vaskuri et al. (2018) includes for each TOC retrieved data point also the calculation of other atmospheric parameters, such as aerosol optical depth and Rayleigh scattering with its parametrizations in the atmospheric model. This procedure of including the relevant atmospheric parameters of the actual existing atmosphere allows the operational use at other locations worldwide with different atmospheric conditions.





Currently, the stratospheric temperature is not retrieved by the algorithm and a climatological averaged value of -45°C is
chosen instead as for the standard Brewer retrieval. The strong seasonal trend correlates with the variation of the stratospheric
temperature at Davos. Effective stratospheric ozone temperatures from ozone launches in Payerne (Switzerland) or from the
ECMWF reanalysis as presented in Gröbner et al. (in review 2021) show a difference of about 10 K between summer time and
wintertime as it is shown for the differences in TOC. This indicates that the algorithm for Koherent may be strongly sensitive
to stratospheric temperature. In a future version of the algorithm we will investigate if stratospheric temperature can be
retrieved from the measurements or be included in the retrieval.

The strong ozone slant path dependency of Koherent in Figure 10 also reflects this seasonal trend of the deviations, since
higher SZA and corresponding higher air-masses are more pronounced in wintertime than in summertime. Figure 10 b) shows
the ozone slant path dependency restricted to one month with an almost constant stratospheric ozone temperature of around
230 K. The graph reveals, that the slant path dependency is significantly lower (-0.0015%/TOC·AMF) than for the entire
period (-0.0033%/TOC·AMF), supporting the conclusion that the strong slant path dependency is caused by biases due to
stratospheric temperature differences.

As mentioned in the Introduction and stated in (Zuber et al., 2018a), the BTS array-spectroradiometer slightly suffer from
straylight at short wavelengths (below 310 nm) at high SZA. A potential effect of this remaining straylight can explain the
increase of the differences at high slant paths above 900 DU (Figure 10 b). However, a proper assessment of the effect of
straylight using the full spectrum least square algorithm between 305 nm and 350 nm, can only be addressed including all
seasons when the effect of stratospheric temperature can be corrected. Based on this dataset, the seasonal trend caused by
stratospheric ozone temperature seems to be much stronger than a potential effect of straylight.

However, when restricted to a summer month with almost constant stratospheric temperature (e.g. July 2020, Figure 10 b))
Koherent shows a slope of the slant path dependency of -0.0015%/TOC·AMF which is comparable to the slant path
dependency between a single and a double monochromator Brewer with a slope of -0.0017%/TOC·AMF.

For BTS Solar, potential straylight impact on TOC is less pronounced. In figure 2, a slight deviation of 5 DU can be observed
in the early morning at high SZA. The statistical analysis of the slant path dependency reveals a linear trend of -
0.00056%/TOC·AMF, which is less than for single monochromator Brewers. Practically the measurement range of TOC is
comparable since also the TOC values of double monochromator Brewer at high SZA/AMF are usually not used due to too
high SNR (e.g. DWD applies an AMF filter >3.5 single monochromator Brewer or >4 double monochromator to TOC values).
We assume that the impact of straylight is less pronounced than for Koherent, because the first wavelength band of the BTS
Solar is an average between 307 nm and 311 nm, where less straylight can be expected, than using the full spectrum until 305
nm for the least square fit algorithm. Most likely the least square fit algorithm of Koherent is more sensitive to biases at short
wavelengths.

The calibration of Koherent showed to be stable over to whole measurement campaign, however an offset was detectable
which we assume is based on the calibration difficulty of a lens-based telescope for an irradiance calibration. The large
variation of 2.2% (*k=1*) of the resulting TOC using the different responsivities obtained by the calibrations highlight the
difficulties of the alignment of the telescope-based system and of an improved calibration is required.

In this respect the irradiance calibration of the BTS Solar seems to be easier applicable, since only a global diffusor system
has to be calibrated using standard irradiance calibration with easily accessible reference plane instead of a lens-based
telescope. The fibre-coupled telescope system has a significant lower light throughput and is therefore more sensitive to noise
at short wavelengths. The signal to noise ratio could eventually also be improved by an optimized alignment of the fibre at the
BTS entrance for maximizing the intensity at short wavelengths.

In the future, a recalibration of Koherent is planned with an improved calibration setup and comparison with additional
reference instruments.



In general, the BTS Solar is comparable to the Brewer 010 with a mean deviation of 0.04% and an expanded standard deviation ($k=2$) of 1,29% and Brewer 226 with a mean deviation of 0.06% and an expanded standard deviation ($k=2$) of 1.47%. The Dobson 104 showed a mean deviation of -0.84% and an expanded standard deviation ($k=2$) of 2.22%.

Koherent achieved a deviation of 1.64% with an expanded standard deviation ($k=2$) of 2.72% by showing a significant seasonal drift of up to 4%.

In the discussion of replacing standard ozone observing instruments, the comparison of Brewers and Dobsons has a long tradition. It was observed that using the Paur and Bass (1985) cross-section, the Brewers and Dobsons show a seasonal difference depending on the stratospheric temperature. Redondas et al. (2014) showed that this difference is significantly smaller when using the IUP cross-section (Serdyuchenko et al., 2013). Recently, Gröbner J. (2021) compared Brewers and Dobsons using other different cross-sections and including stratospheric temperature. The study, especially the data of Koherent, highlights again the importance of applying a specific cross-section and its stratospheric temperature dependency in order to homogenize TOC data from different co-located instruments. From this point of view the new systems in general should be further compared at different locations with different atmospheric conditions and the optimal cross-section for the best homogenization of the data should be systematically investigated.

## 5 Conclusion and Outlook

In summary, the BTS Solar performance regarding TOC determination with data of more than one year is comparable to the performance of the Brewers used in this study. The slant path dependencies (slope of the curve) of the BTS Solar may hint to some remaining straylight at high SZA (above 65°), which is less than a single monochromator Brewer but larger than of a double monochromator Brewer.

The LUT retrieval algorithm used at Hohenpeißenberg was insensitive to stratospheric temperature and provided accurate results by a simple modelling. However a validation in the presence of significant $SO_2$ or aerosol levels could not be tested. The performance in very special stations regarding the atmospheric composition should be further investigated.

Koherent showed a comparable performance to a double monochromator Brewer regarding TOC determination in summer time (around 1% bias), however, with a significant seasonal drift in the winter time of down to -3%. The least square fit algorithm of Koherent is expected to be insensitive to atmospheric changes, due to the included retrieval of atmospheric parameters, such as aerosol optical depth and Rayleigh scattering. Currently, the algorithm of Koherent is sensitive to stratospheric temperature, which explains the seasonal variability relative to the Brewer No. 163. In a future version of the retrieval algorithm, the stratospheric temperature will either be retrieved by the algorithm, or added as an additional parameter to the retrieval routine in order to reduce the observed seasonal variation.

*Competing interests.* The authors declare that they have no conflict of interest.

*Acknowledgement:* The study was partly funded (PMOD/WRC) by the ESA project QA4EO, No. QA4EO/SER/SUB/09 and by the project INFO3RS funded by MeteoSwiss, Grant number 123001926.

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
