# Peer review of "TOC intercomparison of Brewer, Dobson and BTS Solar at Hohenpeißenberg and Davos 2019/2020"

_Atmospheric Measurement Techniques, 2021_

## Referee Comment (RC1)

Review of "TOC intercomparison of Brewer, Dobson and BTS Solar at Hohenpeißenberg and Davos 2019/2020"

Vladimir Savastiouk

This is an important contribution to the continuing efforts for expanding our capabilities in monitoring the ozone layer. The paper describes a TOC intercomparison using the well-established Brewer and Dobson ozone spectrophotometers together with newer BTS array spectrometers. The description of the intercomparison is sufficiently detailed and the conclusion contains important steps for further improvement of the new instruments and the retrieval algorithms. The results of the intercomparison are encouraging.

There are some important shortcomings in the current state of the paper. These are mostly form-related, but some are content-related as well.

First, this is likely the longest Introduction I've seen in a such a short paper. I highly recommend cutting it in half. The long list of which reference paper describes which instrument is likely unnecessary.

Also to this, an inappropriately detailed description of the Dobson spectrophotometer is out of place in this paper, especially when an exhaustive reference list is provided.

The authors keep referring to the array-based measurements as "continuous spectral range" and contrast this with the "discrete wavelngths" type of the Brewer and the Dobson. I truly dislike such terminology since the only difference, however important, is in the number of the wavelengths. There is no way to either record or analyze "continuous spectral range". I recommend to either define what you call "continuous spectral range" or not use this term.

In lines 142-143 the paper incorrectly states that only one wavelength is used for SO2. In fact 5 wavelengths are used for SO2.

Lines 145-150 have a somewhat confusing discussion about the time needed for a measurement in different instruments. The discussion seem to first suggest that both the Brewers and the Dobsons take too long compared to BTS only to finish by saying the indeed it takes up o 5 min for BTS to collect good statistics. I recommend to either express this though clearly as to why you see this important or remove this from the paper.

Lines 239-241 must be re-written to a) correctly define what 'm' is and b) to explain how it is possible to have same AMF for ozone,aerosol and Rayleigh (it isn't).

Lines 247-248 may need a more accurate statement about shy it is possible to retrieve Rayleigh because it is definitely not due to "advantage of the minimal least square fit". Hint: if Rayleigh were to correlate with ozone the retrieval would fail.

Line 380 may lead the readers to conclude that the strong seasonal trend is somehow related to the Brewers. Please clarify/re-phrase.

I recommend to re-work the flow of lines 389-395 to have a more logical order of the discussion of the straylight and its effect on the seasonality in the differences.

Lines 404-405. Assume it's a typo: "too high" meant to be "too low"?

This is important: almost all figures use a colour scheme that is poor for presentation. Please use more contrasting colours for different lines/points. Also in figures: some lines are only marked as "fit" while no explanation is found how those fit were done.

Cosmetic corrections:

line 12: "fibre-coupled", "optics", "optics"
line 21: consider re-wording "the slant path slope" or define what you mean
line 25: "is" instead of "has been"
line 107: way too many decimal points for the lat/lon.
line 245: re-word "parametrized with a linear parametrization"
line 414: "applied" instead of "applicable"
line 416: "significantly"

---

## Author Comment (AC1)

**Authors response to comments:**

**The authors thank Vladimir Savastiouk for the detailed review and comments. See our response and corrections in order to improve the publication:**

Review of "TOC intercomparison of Brewer, Dobson and BTS Solar at Hohenpeißenberg and Davos 2019/2020"
Vladimir Savastiouk

This is an important contribution to the continuing efforts for expanding our capabilities in monitoring the ozone layer. The paper describes a TOC intercomparison using the well-established Brewer and Dobson ozone spectrophotometers together with newer BTS array spectrometers. The description of the intercomparison is sufficiently detailed and the conclusion contains important steps for further improvement of the new instruments and the retrieval algorithms. The results of the intercomparison are encouraging.
There are some important shortcomings in the current state of the paper. These are mostly form related, but some are content-related as well.

| Comment | Authors response |
|---|---|
| First, this is likely the longest Introduction I've seen in a such a short paper. I highly recommend cutting it in half. The long list of which reference paper describes which instrument is likely unnecessary. | We think this introduction is helpful since a new type of device is introduced in a long term intercomparisons. This is why literature to established systems might be helpful for some readers. We shortened one section and moved one section into 2.1. |
| Also to this, an inappropriately detailed description of the Dobson spectrophotometer is out of place in this paper, especially when an exhaustive reference list is provided. | |
| The authors keep referring to the array-based measurements as "continuous spectral range" and contrast this with the "discrete wavelngths" type of the Brewer and the Dobson. I truly dislike such terminology since the only difference, however important, is in the number of the wavelengths. There is no way to either record or analyze "continuous spectral range". I recommend to either define what you call "continuous spectral range" or not use this term. | We agree that continuous is not the exact expression. We have removed this from the manuscript. We further have clarified and defined the meaning of the expression "full spectrum" in order to distinguish in one expression from the ozone retrieval from the Brewer or Dobson wavelengths. |
| In lines 142-143 the paper incorrectly states that only one wavelength is used for SO2. In fact 5 wavelengths are used for SO2. | Accepted |
| Lines 145-150 have a somewhat confusing discussion about the time needed for a measurement in different instruments. The discussion seem to first suggest that both the Brewers and the Dobsons take too long compared to BTS only to finish by saying the indeed it takes up o 5 min for BTS to collect good statistics. I recommend to either express this though clearly as to why you see this important or remove this from the paper. | We modified this section in order to express the capability and our considerations more. |
| Lines 239-241 must be re-written to a) correctly define what 'm' is and b) to explain how it is possible to have same AMF for ozone, aerosol and Rayleigh (it isn't). | Thank you for this important comment. We agree that the air masses are different for aerosol, Rayleigh and ozone. We have specified in more detail the method used for the retrieval of the presented data. We have clarified this in the revised manuscript. We have specifically written the composition of air mass m in Eq. 2. For ozone, aerosol and Rayleigh, separately (Eq. 2). |
| Lines 247-248 may need a more accurate statement about shy it is possible to retrieve Rayleigh because it is definitely not due to "advantage of the minimal least square fit". Hint: if Rayleigh were to correlate with ozone the retrieval would fail. | We have addressed this comment in the revised manuscript to clarify that only ozone and aerosol are used as fitting parameters of the least square fit. We also highlighted that these parameters are weakly correlated. We agree that correlations would not allow using the minimal least square fit approach.
Furthermore, we clarified now that Rayleigh is not retrieved, but used as a parametrization to model the atmosphere. |
| Line 380 may lead the readers to conclude that the strong seasonal trend is somehow related to the Brewers. Please clarify/re-phrase. | Corrected by removing the relation to the Brewers. |

| | |
|---|---|
| I recommend to re-work the flow of lines 389-395 to have a more logical order of the discussion of the straylight and its effect on the seasonality in the differences. | For better understanding, we have better structured this section in the revised manuscript |
| Lines 404-405. Assume it's a typo: "too high" meant to be "too low"? | Yes, we corrected this typo. |
| This is important: almost all figures use a colour scheme that is poor for presentation. Please use more contrasting colours for different lines/points. Also in figures: some lines are only marked as"fit" while no explanation is found how those fit were done. | We agree that figure 6 is not optimal in the color scheme. We adapted this figure. All others seem appropriate. The fits are described in the text or subtitle. |
| **Cosmetic corrections:** | |
| line 12: "fibre-coupled", "optics", "optics" | Accepted |
| line 21: consider re-wording "the slant path slope" or define what you mean | In the abstract no definition is needed. We added a small definition in line 300. |
| line 25: "is" instead of "has been" | Accepted |
| line 107: way too many decimal points for the lat/lon. | Accepted |
| line 245: re-word "parametrized with a linear parametrization" | Accepted |
| line 414: "applied" instead of "applicable" | Accepted |
| line 416: "significantly" | Accepted |

---

## Author Comment (AC2)

**The authors thank the anonymous referee for the detailed review and comments. See our response and corrections in order to improve the publication:**

This is the first review of the paper submitted to AMTD by R. Zuber et al. The paper is titled "TOC intercomparison of Brewer, Dobson and BTS Solar at Hohenpeißenberg and Davos 2019/2020" and is focused on discussion of the BTS instrumental performance with different optical system setups at two established ground-based stations in Europe. The authors address the benefits and limitations of the new instrument and two algorithms used to process the data. Comparisons against one Dobson and several Brewer coincident observations are discussed in the paper. The authors discuss stray light interference and temperature sensitivity in the BTS-derived total column ozone. Results of comparisons are of interest to the ozone community to understand biases and seasonal dependencies in the established and new ozone observing systems. With the advancement of the geostationary satellite observing systems and the societal focus on understanding air quality impacts on human health and the environment, the high temporal resolution in ozone observations that can provide high accuracy and stability offer support for monitoring ozone changes in the range of minute to seasonal scales and with a hands-off approach. The authors acknowledge the need for future improvements in the data processing and improved modeling of observations instead of look-up tables.

This paper is structured well, addressing various aspects of comparisons. One would wish the authors had a longer period of data at both stations to address seasonal variability. Also, data processing and optical system differences make comparisons and conclusions complicated. Ideally, it would be great to have BTS Solar and Coherent observations done at the same location to compare the performance of both systems and a setup. On the other hand, Hohenpeißenberg and Davos are located at a close distance from each other, and all Brewers have been recently calibrated and therefore should be performing similarly at both locations. Therefore, I would recommend accepting this paper for publication after all comments are answered.

➔ Comment: We agree to have Koherent and BTS Solar for a longer time at one station would be good. We keep this in mind for our future considerations. For this intercomparison this was not possible.

I would recommend that the authors ask for help from an English-speaking colleague to improve the readability of the text.

The authors use the terminology "expanded standard deviation". If it is the same as 2 standard deviations, please add this explanation in the text (or refer to 95 % confidence limits). ➔ k=2 is added.

Detailed comments: ("accepted" means we corrected the manuscript accordingly)

| | |
|---|---|
| Lines 14:15. "The array-spectrometer-based BTS systems have been **traceable calibrated** to National Metrology Institutes (NMI) and the used TOC retrieval algorithms" – you should choose either traceable or calibrated. Instead of "used" select "respected" or "both versions of". | It is called traceable calibrated, we kept this. We accepted the second suggestion. |
| Line 16: add "wavelength pair for Dobson" as Dobson does not measure at individual wavelengths (as you discuss later in the text). | Accepted |
| Line 18 "deviation of the Solar BTS and Brewer" – did you mean difference from Brewer total column ozone? | Accepted |
| Line 19 "deviation" – is it mean bias or standard deviation (one sigma)? You can replace "given" with "caused". | Expanded standard deviation is understood as k=2. We added this to make it clearer. |
| Line 20 – is it continuous drift or seasonal bias? | Accepted |
| Consider re-writing the sentences starting from "Resulting", here is one option:

To summarize, the BTS Solar instrument performed at the level of Brewer stability and accuracy during the intercomparison campaign held in Hohenpeissenberg, Germany in 2019/2020." | Accepted |
| Line 25 "defined" -> "recognized" | Accepted |
| Line 30 "bit no further decline either" -> was either observed? | Accepted |
| Line 32 "monitoring of the protocol for the CFC ban" -> monitoring protocol for banned CFCs? | Accepted |
| Line 35 "argument why further observations will be necessary" -> "requirement for continuing observations". | Accepted |
| Line 37 "when the at that time" -> with the development of the Dobson, built by | Accepted |
| Line 38 "A first small" -> "The first small" | Accepted |
| Line 50 "Publications about the function of Brewer spectrometers" -> "Publications describing the Brewer spectrophotometer" | Accepted |

| | |
|---|---|
| Line 57 "newly" -> recently? | Accepted |
| Line 65 – (2 and 2x2 wavelengths)? should it be "single or double pair observations" | Accepted |
| Line 66 "It is expected that this additional " – Do you have a reference to the paper? | Accepted: Since there is no reference for this assumption we re-worded to "One may assume that… |
| Line 69 "within an intercomparison" -> at the intercomparison campaign and reported by Egli et al., 2016 | Accepted |
| Line 73 "range of 5 %"  - is this error used for the irradiance or total ozone results? If it is for total ozone, then why is 5 % acceptable and not 1 %, which is the goal for direct sun observations at higher SZAs? If the instrument measures poorly at large SZAs, why use it? | Solar Irradiance added. We did not use the array spectroradiometers from the mentioned comparison. The used BTS presented here was not part of that paper. See line 88. |
| Lines 79 and 80. Please make it clear that Dobson was not corrected for artifacts of the stray light. Moreover, only AD-pair direct sun Dobson observations were used in comparisons with Pandora in Boulder, CO that were taken within the acceptable range of air masses that would minimize the impact of stray light observations. | Thanks for clarification. We have revised the manuscript accordingly |
| Line 86 "released" -> developed? "quality assessment" -> "assessment of quality" | Release is correct, the development took already place at that time. Second comment accepted. |
| Line 87-88 "The BTS …. In terms of solar global spectral irradiance" -> "The accuracy and stability of the BTS's solar global spectral irradiance were compared against the  well-established double monochromator-based systems, such as double Brewer and ?" | Accepted and slightly modified |
| Line 92 "wavelength" used twice in the sentence | Accepted |
| Line 103 "long term" – define how long, i.e. 3 months, one year… | Accepted |
| Line 111 "belong as" -> is part of | Accepted |
| Line 114 "double Brewer #163"? | Accepted |
| Line 137 define "very good calibration-level", please be more specific | Accepted |
| Lines 146-150 – if this discussion was to show the advantage of the BTS for faster observations than available in Brewer schedule, it failed after I read the following statement "however usually an averaging of 1 to 5 min is applied" which is similar to 3-min for Brewer integration time. Please modify this section. | We expressed that usually this averaging is done in order to reduce the amount of data and optimize the SNR. Furthermore, we rephrased the paragraph a bit to express more clearly the intention. |
| Line 160 "in principle a full least square algorithm" – not clear what you are trying to say. The least-square fit to the spectral observations is used to derive TOC? Or "the TOC algorithm is based on the least square fit in the spectral range of 305-350 nm" | Accepted and revised the manuscript. |
| Line 162 "validate"? Do you mean test or reduce? | We mean validate. We corrected this sentence since it was misleading. |
| Line 175 "dynamic" -< variability? | Accepted |
| Line 176 "maximum 2.5 DU" – but just before that statement, the error is claimed to be <0.8 DU. | Very good comment. This sentence was wrong. We rewrote it. |
| Line 196 Yyou are using the climatological profiles embedded in the Libtran software to derive the total ozone column from BTS observations. Since the shape of the profile becomes more important at large SZAs, have you compared standard profiles against the ozonesonde record of Hohenpeissenberg to prove that these profiles are representative and do not introduce additional errors? In addition, you are using 22 km to derive the airmass factor. How does it compare with the Libtran ozone profile shape? | The aim was to use this crude modelling in order to show that it is already precise enough. Of course a more detailed modelling would improve it even more. However we wanted to show that this is sufficient in Hohenpeißenberg, what makes the application of the algorithm easier. We expressed this in this chapter, but especially also see Zuber et al. (2018b). |
| Line 200 – Does this statement hold for TOC at large SZAs? | We compared the diurnal plot and could not see significant differences in Hohenpeißenberg within this intercomparison at the considered AMF. Short phrase added to manuscript. |
| Line 213 and again on line 223. How did you select 10 DU as a quality criterion? | As stated in the sentence: "Since such a large change in TOC within such a short time interval can only be expected due to instrument malfunction, or cloud movement or very high SZA." We used this value since it is significantly larger than the measurement uncertainty and difference which can be expected in such a short time difference. |
| Line 219. What is the field of view for the BTS Solar and how does it compare with the Koherent field of view? | Koherent is given with +/- 0.6°, we added the FOV of the BTS Solar with +/- 1.4°. This is given in the cited Zuber et al. (2018b) |

| | |
|---|---|
| Line 234  It could help to introduce an abbreviation for the "least squares algorithm" throughout the paper after you first introduced it. | We think an abbreviation might be possible but not needed. We remain it as it is. |
| Line 268 "additionally part"? Do you mean "additional observations during intercomparisons" Or special observations? Please explain. | Accepted |
| Line 274 "Exemplary" – are these truly "the best days of the entire field campaign"? Or did you mean "examples of daily variability in TOC observations"? | These are not the best days. These are just two examples which show a strong diurnal dynamic as stated. Slightly rephrased. |
| Line 276 Did you mean "capture the same TOC variability with time/SZA"? | Accepted |
| "winter times" -> "winter season" I also see that Dobson was able to capture the diurnal variability of July 9th observations shown in Figure 3, right panel. Although Dobson does not provide continuous observations, it is quite capable of capturing atmospheric changes. Please include this information in the text. | Accepted. |
| Also, in the legend on the right, the mean ozone value for Dobson is 308 DU. However, based on the data shown in the plot, it seems to be the wrong number – please check. | We refer to the information to the legend. We are considering data between 10:00 to 12:00. |
| Also, is it correct that Dobson's observations on July 7th started before 8 am? What was this type of observation, probably not AD direct sun? Dobson data are typically reported in local time. How was the conversion to the UTC done? | Yes, I assume April 7th is meant. The Dobson measurements start at 7:29 CET at a mue-value of 3.24, which is sufficient for AD observations. |
| You should also add the uncertainty of each observation to the plots to show how different products compare. | Currently the absolute uncertainty of Brewers and Dobsons are not known and can therefore not be marked with error bars. The agreement of the Brewers and Dobsons are within 1% compared to the reference. We have added a citation regarding Brewers (Redondas et al. 2019) |
| Line 290 or part of Figure caption: "a worse performance" – why was the Dobson instrument's worse performance? | Rephrased |
| Line 293 "trends"-> results | Accepted |
| Line 294 " the least square fit is within 1 % over the whole measurement campaign" – Are you saying that every spectral fit was within 1 % of the observed spectrum or you are saying that the retrieval method that uses the LSF derived the TOC that was within 1 % of the Brewer-derived TOC? | We are saying that the fit of the plot stays within +/- 1% over the whole measurement campaign. We tried to make it clearer that this refers to the figure. |
| Figure 5 – why is the range of the individual differences (black squares) between Dobson and BTS is small in comparison to the Brewer/BTS comparisons (large spread in blue and green squares)? | This can be explained by the fact that BTS and Brewer deliver much more data points on each day, even for not ideal weather conditions (clouds, higher SZA, etc.) |
| This brings the question about the results shown in Figure 4. Does the histogram include the seasonal bias? | It includes all effects, so yes. |
| I wonder if you remove the seasonal bias (correct Dobson for the effective temperature bias) and repeat the histogram would the Gaussian shape be as wide? | We can assume that correction would improve the results. For this study we did not correct neither the Dobson or Brewer for stratospheric temperature. Redondas et al. 2014 addressed this question and Gröbner et al. 2021 recently presented Brewer and Dobson data including the stratospheric temperature effect correction (we added a citation). |
| Line 325 "percentual" -> percent? | Accepted |
| "overestimation of Koherent of a mean" ->" overestimation by Koherent on average by 1.64%" | Accepted |
| Line 327 "in the order as for" -> "comparable to" | Accepted |
| Figure 9: Histogram shows two distributions and incorporates the seasonal offset.  It is better to show comparisons for each season separately, similarly to what you are doing in Figure 10. | Yes, that would be an appropriate solution too. However, we wanted to show how it performs over the seasons without any stratospheric temperature correction |
| Line 323 "evidenced their performance" -> demonstrated instrument performance | Line 352: Accepted |
| Line 364 "simple modeling" – it would be useful to test the sensitivity of both TOC retrieval algorithms to the ozone profile shape. Most of the TOC retrievals (except in Antarctica during the spring ozone depletion) are not sensitive to the vertical ozone distributions except at large SZAs. | This could be done in further research. We thank for this suggestion, but this analysis exceeds the scope of this paper. |
| Line 370 change the to The at the beginning of the sentence | Accepted |

| | |
|---|---|
| Line 377 "relevant atmospheric parameters" – explain what you mean. Are you saying that the retrieval will be improved if aerosols and SO2 information would be available to constrain the spectral fitting? | The inclusion of measured aerosols or SO2 as input parameters was not investigated. Aerosols and Rayleigh are free fitting parameters of the least square fit. We have rephrased this part. |
| Line 378 – "actual atmosphere" -> observed atmosphere | See above. We have rephrased this part. |
| Line 387 "higher latitude"? | It should be higher SZA as in the original version of the manuscript. |
| Line 392 define "slightly" | This word is removed in the revised manuscript |
| Line 402 "linear trend"-> slope | Accepted |
| Line 404 "too high" – please define | See definition int the original manuscript in the brackets and this further explanation. Long term experience revealed that the single Brewer TOC drops already at an average AMF > 3.5 due to stray light effects, whereas a double Brewer with better stray light suppression is able to measure reliable TOC up to AMF = 4. We added a sentence to connect to this information. |
| Line 411 what do you mean by "calibration difficulty"? Please rephrase. | Accepted and rephrased |
| Line 425 and therefore comparable to Dobson? | Since we did not compare Koherent with Dobsons in Davos we cannot reliably cover such a statement. |
| I did not find information on where the data from these observational campaigns are archived or how these data can be obtained. | The data is available from: https://doi.org/10.6084/m9.figshare.14686656

We have added this information in the revised manuscript. |

---

## Author Response (AR2)

**Authors response to comments:**

In general, we really appreciate the review comments since it improved the paper significantly. We tried to express this also in the acknowledgements section. In addition, linguistic corrections were made by a native speaker.

**The authors thank Natalya Kramarovafor the detailed review and comments. See our response and corrections in order to improve the publication:**

Comments to the Author:
Dear Authors,

Thank you for addressing referees' comments and uploading the revised manuscript. It reads much better after the revision; however, I still think the manuscript will benefit from another round of proofreading. There are some parts of the manuscript that are hard to understand, mostly because of the language. Below I listed several suggestions that I would like you to address (these notes are applicable to the file with authors tracking changes):

➔ The manuscript has been reviewed by a native speaker.

| | |
|---|---|
| Abstract, line 8: replace "In the 2019/2020 measurement campaign" with "During the 2019/2020 measurement campaign";
 Abstract, line 19: replace "within the whole measurement campaign" with "over the whole measurement campaign";
 Abstract, line 21: replace "at the comparison in Hohenpeißenberg" with "in the comparisons at Hohenpeißenberg";
 Abstract, line 22: It is not clear to me what you are saying here "The slant path slope is in-between double monochromator and single monochromator Brewer". I guess you are talking about results shown in Fig. 6. Then it would be better to re-phrase it, something like "The BTS Solar had shown very small dependence on the slant path column compared to the double monochromator Brewer and performed better than the single monochromator Brewer."
 Page 1, lines 36-37: replace with "… is another argument for continuing observations."
 Page 2, line 62: since you modified this section, it is not clear from the context what do you mean by "such new instruments". Perhaps, you can say "State-of-the-art array spectrophotometers measuring spectral UV radiation are potential candidates for the new types of instruments to monitor the ozone layer". And later replace "These have e.g." with "These instruments have an advantage …"
 Page 2, line 67: replace "wavelength values" with "wavelengths";
 Page 2, line 69: it might be better to replace this phrase with "Measurements obtained at multiple wavelengths with a narrow spectral step we will call here as …"
 Page 5, line 158: it would be better to say "…at each of the five wavelengths used for ozone and…"
 Page 6, line 168: I propose to replace that with "… to collect a reasonable amount of data and optimize …"
 Page 7, line 240: is it really "70 cm distance"?
 Page 8, lines 279-280: the text now reads like you used different atmospheric profiles to calculate ozone, Rayleigh and aerosol optical thicknesses for the same measurement. But I guess that is not what you meant here. It might be better to change the text as "The airmasses for the ozone, aerosol and Rayleigh optical thickness (Eq. 2) are calculated independently for various profiles from the standard US atmosphere …"
 Page 9, line 286: should be "The assumptions of -45°C is the same as used in the standard Brewer procedure."
 Page 9, line 300: do not remove "resulting" here as it would change the meaning.
 Figure 3, figure caption: In the second sentence it would be better to say "Measurements on 07.04.2020 show a strong dynamically-driven change in TOC within the day"
 Figure 4, figure caption: the figure shows the histogram of deviations of BTS Solar from Brewer No. 010, Brewer No. 226 and Dobson No. 104.
 Figure 5, figure caption: TOC differences between BTS Solar and Brewers/Dobson over the whole campaign at Hohenpeißenberg.
 Figure 6, figure caption: TOC differences as a function of slant path column between BTS Solar and Brewers at Hohenpeißenberg.
 Page 16, line 474: should this be "seems to be easier to apply"?
 Page 16, line 483: Are you quoting here the deviation of Dobson 104 from BTS Solar or from Brewers? Please, clarify. | All accepted and corrected or clarified accordingly. |

| Page 16, line 484: Are you talking about differences between Koherent and Brewer163? Please, specify. | |

**The authors thank Vladimir Savastiouk for the detailed review and comments. See our response and corrections in order to improve the publication:**

Review of "TOC intercomparison of Brewer, Dobson and BTS Solar at Hohenpeißenberg and Davos 2019/2020"
Vladimir Savastiouk

This is an important contribution to the continuing efforts for expanding our capabilities in monitoring the ozone layer. The paper describes a TOC intercomparison using the well-established Brewer and Dobson ozone spectrophotometers together with newer BTS array spectrometers. The description of the intercomparison is sufficiently detailed and the conclusion contains important steps for further improvement of the new instruments and the retrieval algorithms. The results of the intercomparison are encouraging.
There are some important shortcomings in the current state of the paper. These are mostly form related, but some are content-related as well.

| Comment | Authors response |
|---|---|
| First, this is likely the longest Introduction I've seen in a such a short paper. I highly recommend cutting it in half. The long list of which reference paper describes which instrument is likely unnecessary. | We think this introduction is helpful since a new type of device is introduced in a long term intercomparisons. This is why literature to established systems might be helpful for some readers. We shortened one section and moved one section into 2.1. |
| Also to this, an inappropriately detailed description of the Dobson spectrophotometer is out of place in this paper, especially when an exhaustive reference list is provided. | |
| The authors keep referring to the array-based measurements as "continuous spectral range" and contrast this with the "discrete wavelngths" type of the Brewer and the Dobson. I truly dislike such terminology since the only difference, however important, is in the number of the wavelengths. There is no way to either record or analyze "continuous spectral range". I recommend to either define what you call "continuous spectral range" or not use this term. | We agree that continuous is not the exact expression. We have removed this from the manuscript. We further have clarified and defined the meaning of the expression "full spectrum" in order to distinguish in one expression from the ozone retrieval from the Brewer or Dobson wavelengths. |
| In lines 142-143 the paper incorrectly states that only one wavelength is used for SO2. In fact 5 wavelengths are used for SO2. | Accepted |
| Lines 145-150 have a somewhat confusing discussion about the time needed for a measurement in different instruments. The discussion seem to first suggest that both the Brewers and the Dobsons take too long compared to BTS only to finish by saying the indeed it takes up o 5 min for BTS to collect good statistics. I recommend to either express this though clearly as to why you see this important or remove this from the paper. | We modified this section in order to express the capability and our considerations more. |
| Lines 239-241 must be re-written to a) correctly define what 'm' is and b) to explain how it is possible to have same AMF for ozone, aerosol and Rayleigh (it isn't). | Thank you for this important comment. We agree that the air masses are different for aerosol, Rayleigh and ozone. We have specified in more detail the method used for the retrieval of the presented data. We have clarified this in the revised manuscript. We have specifically written the composition of air mass m in Eq. 2. For ozone, aerosol and Rayleigh, separately (Eq. 2). |
| Lines 247-248 may need a more accurate statement about shy it is possible to retrieve Rayleigh because it is definitely not due to "advantage of the minimal least square fit". Hint: if Rayleigh were to correlate with ozone the retrieval would fail. | We have addressed this comment in the revised manuscript to clarify that only ozone and aerosol are used as fitting parameters of the least square fit. We also highlighted that these parameters are weakly correlated. We agree that correlations would not allow using the minimal least square fit approach.
Furthermore, we clarified now that Rayleigh is not retrieved, but used as a parametrization to model the atmosphere. |
| Line 380 may lead the readers to conclude that the strong seasonal trend is somehow related to the Brewers. Please clarify/re-phrase. | Corrected by removing the relation to the Brewers. |
| I recommend to re-work the flow of lines 389-395 to have a more logical order of the discussion of the straylight and its effect on the seasonality in the differences. | For better understanding, we have better structured this section in the revised manuscript |
| Lines 404-405. Assume it's a typo: "too high" meant to be "too low"? | Yes, we corrected this typo. |
| This is important: almost all figures use a colour scheme that is poor for presentation. Please use more contrasting colours for | We agree that figure 6 is not optimal in the color scheme. We adapted this figure. All others seem |

| | |
|---|---|
| different lines/points. Also in figures: some lines are only marked as"fit" while no explanation is found how those fit were done. | appropriate. The fits are described in the text or subtitle. |
| **Cosmetic corrections:** | |
| line 12: "fibre-coupled", "optics", "optics" | Accepted |
| line 21: consider re-wording "the slant path slope" or define what you mean | In the abstract no definition is needed. We added a small definition in line 300. |
| line 25: "is" instead of "has been" | Accepted |
| line 107: way too many decimal points for the lat/lon. | Accepted |
| line 245: re-word "parametrized with a linear parametrization" | Accepted |
| line 414: "applied" instead of "applicable" | Accepted |
| line 416: "significantly" | Accepted |

**The authors thank the anonymous referee for the detailed review and comments. See our response and corrections in order to improve the publication:**

This is the first review of the paper submitted to AMTD by R. Zuber et al. The paper is titled "TOC intercomparison of Brewer, Dobson and BTS Solar at Hohenpeißenberg and Davos 2019/2020" and is focused on discussion of the BTS instrumental performance with different optical system setups at two established ground-based stations in Europe. The authors address the benefits and limitations of the new instrument and two algorithms used to process the data. Comparisons against one Dobson and several Brewer coincident observations are discussed in the paper. The authors discuss stray light interference and temperature sensitivity in the BTS-derived total column ozone. Results of comparisons are of interest to the ozone community to understand biases and seasonal dependencies in the established and new ozone observing systems. With the advancement of the geostationary satellite observing systems and the societal focus on understanding air quality impacts on human health and the environment, the high temporal resolution in ozone observations that can provide high accuracy and stability offer support for monitoring ozone changes in the range of minute to seasonal scales and with a hands-off approach. The authors acknowledge the need for future improvements in the data processing and improved modeling of observations instead of look-up tables.

This paper is structured well, addressing various aspects of comparisons. One would wish the authors had a longer period of data at both stations to address seasonal variability. Also, data processing and optical system differences make comparisons and conclusions complicated. Ideally, it would be great to have BTS Solar and Coherent observations done at the same location to compare the performance of both systems and a setup. On the other hand, Hohenpeißenberg and Davos are located at a close distance from each other, and all Brewers have been recently calibrated and therefore should be performing similarly at both locations. Therefore, I would recommend accepting this paper for publication after all comments are answered.

➔ Comment: We agree to have Koherent and BTS Solar for a longer time at one station would be good. We keep this in mind for our future considerations. For this intercomparison this was not possible.

I would recommend that the authors ask for help from an English-speaking colleague to improve the readability of the text.

The authors use the terminology "expanded standard deviation". If it is the same as 2 standard deviations, please add this explanation in the text (or refer to 95 % confidence limits). ➔ k=2 is added.

Detailed comments: ("accepted" means we corrected the manuscript accordingly)

| | |
|---|---|
| Lines 14:15. "The array-spectrometer-based BTS systems have been **traceable calibrated** to National Metrology Institutes (NMI) and the used TOC retrieval algorithms" – you should choose either traceable or calibrated. Instead of "used" select "respected" or "both versions of". | It is called traceable calibrated, we kept this. We accepted the second suggestion. |
| Line 16: add "wavelength pair for Dobson" as Dobson does not measure at individual wavelengths (as you discuss later in the text). | Accepted |
| Line 18 "deviation of the Solar BTS and Brewer" – did you mean difference from Brewer total column ozone? | Accepted |
| Line 19 "deviation" – is it mean bias or standard deviation (one sigma)? You can replace "given" with "caused". | Expanded standard deviation is understood as k=2. We added this to make it clearer. |
| Line 20 – is it continuous drift or seasonal bias? | Accepted |
| Consider re-writing the sentences starting from "Resulting", here is one option:

To summarize, the BTS Solar instrument performed at the level of Brewer stability and accuracy during the intercomparison campaign held in Hohenpeissenberg, Germany in 2019/2020." | Accepted |
| Line 25 "defined" -> "recognized" | Accepted |
| Line 30 "bit no further decline either" -> was either observed? | Accepted |
| Line 32 "monitoring of the protocol for the CFC ban" -> monitoring protocol for banned CFCs? | Accepted |
| Line 35 "argument why further observations will be necessary" -> "requirement for continuing observations". | Accepted |
| Line 37 "when the at that time" -> with the development of the Dobson, built by | Accepted |
| Line 38 "A first small" -> "The first small" | Accepted |
| Line 50 "Publications about the function of Brewer spectrometers" -> "Publications describing the Brewer spectrophotometer" | Accepted |

| | |
|---|---|
| Line 57 "newly" -> recently? | Accepted |
| Line 65 – (2 and 2x2 wavelengths)? should it be "single or double pair observations" | Accepted |
| Line 66 "It is expected that this additional " – Do you have a reference to the paper? | Accepted: Since there is no reference for this assumption we re-worded to "One may assume that… |
| Line 69 "within an intercomparison" -> at the intercomparison campaign and reported by Egli et al., 2016 | Accepted |
| Line 73 "range of 5 %" - is this error used for the irradiance or total ozone results? If it is for total ozone, then why is 5 % acceptable and not 1 %, which is the goal for direct sun observations at higher SZAs? If the instrument measures poorly at large SZAs, why use it? | Solar Irradiance added. We did not use the array spectroradiometers from the mentioned comparison. The used BTS presented here was not part of that paper. See line 88. |
| Lines 79 and 80. Please make it clear that Dobson was not corrected for artifacts of the stray light. Moreover, only AD-pair direct sun Dobson observations were used in comparisons with Pandora in Boulder, CO that were taken within the acceptable range of air masses that would minimize the impact of stray light observations. | Thanks for clarification. We have revised the manuscript accordingly |
| Line 86 "released" -> developed? "quality assessment" -> "assessment of quality" | Release is correct, the development took already place at that time. Second comment accepted. |
| Line 87-88 "The BTS …. In terms of solar global spectral irradiance" -> "The accuracy and stability of the BTS's solar global spectral irradiance were compared against the well-established double monochromator-based systems, such as double Brewer and ?" | Accepted and slightly modified |
| Line 92 "wavelength" used twice in the sentence | Accepted |
| Line 103 "long term" – define how long, i.e. 3 months, one year… | Accepted |
| Line 111 "belong as" -> is part of | Accepted |
| Line 114 "double Brewer #163"? | Accepted |
| Line 137 define "very good calibration-level", please be more specific | Accepted |
| Lines 146-150 – if this discussion was to show the advantage of the BTS for faster observations than available in Brewer schedule, it failed after I read the following statement "however usually an averaging of 1 to 5 min is applied" which is similar to 3-min for Brewer integration time. Please modify this section. | We expressed that usually this averaging is done in order to reduce the amount of data and optimize the SNR. Furthermore, we rephrased the paragraph a bit to express more clearly the intention. |
| Line 160 "in principle a full least square algorithm" – not clear what you are trying to say. The least-square fit to the spectral observations is used to derive TOC? Or "the TOC algorithm is based on the least square fit in the spectral range of 305-350 nm" | Accepted and revised the manuscript. |
| Line 162 "validate"? Do you mean test or reduce? | We mean validate. We corrected this sentence since it was misleading. |
| Line 175 "dynamic" -< variability? | Accepted |
| Line 176 "maximum 2.5 DU" – but just before that statement, the error is claimed to be <0.8 DU. | Very good comment. This sentence was wrong. We rewrote it. |
| Line 196 Yyou are using the climatological profiles embedded in the Libtran software to derive the total ozone column from BTS observations. Since the shape of the profile becomes more important at large SZAs, have you compared standard profiles against the ozonesonde record of Hohenpeissenberg to prove that these profiles are representative and do not introduce additional errors? In addition, you are using 22 km to derive the airmass factor. How does it compare with the Libtran ozone profile shape? | The aim was to use this crude modelling in order to show that it is already precise enough. Of course a more detailed modelling would improve it even more. However we wanted to show that this is sufficient in Hohenpeißenberg, what makes the application of the algorithm easier. We expressed this in this chapter, but especially also see Zuber et al. (2018b). |
| Line 200 – Does this statement hold for TOC at large SZAs? | We compared the diurnal plot and could not see significant differences in Hohenpeißenberg within this intercomparison at the considered AMF. Short phrase added to manuscript. |
| Line 213 and again on line 223. How did you select 10 DU as a quality criterion? | As stated in the sentence: "Since such a large change in TOC within such a short time interval can only be expected due to instrument malfunction, or cloud movement or very high SZA." We used this value since it is significantly larger than the measurement uncertainty and difference which can be expected in such a short time difference. |
| Line 219. What is the field of view for the BTS Solar and how does it compare with the Koherent field of view? | Koherent is given with +/- 0.6°, we added the FOV of the BTS Solar with +/- 1.4°. This is given in the cited Zuber et al. (2018b) |

| | |
|---|---|
| Line 234  It could help to introduce an abbreviation for the "least squares algorithm" throughout the paper after you first introduced it. | We think an abbreviation might be possible but not needed. We remain it as it is. |
| Line 268 "additionally part"? Do you mean "additional observations during intercomparisons" Or special observations? Please explain. | Accepted |
| Line 274 "Exemplary" – are these truly "the best days of the entire field campaign"? Or did you mean "examples of daily variability in TOC observations"? | These are not the best days. These are just two examples which show a strong diurnal dynamic as stated. Slightly rephrased. |
| Line 276 Did you mean "capture the same TOC variability with time/SZA"? | Accepted |
| "winter times" -> "winter season" I also see that Dobson was able to capture the diurnal variability of July 9$^{th}$ observations shown in Figure 3, right panel. Although Dobson does not provide continuous observations, it is quite capable of capturing atmospheric changes. Please include this information in the text. | Accepted. |
| Also, in the legend on the right, the mean ozone value for Dobson is 308 DU. However, based on the data shown in the plot, it seems to be the wrong number – please check. | We refer to the information to the legend. We are considering data between 10:00 to 12:00. |
| Also, is it correct that Dobson's observations on July 7$^{th}$ started before 8 am? What was this type of observation, probably not AD direct sun? Dobson data are typically reported in local time. How was the conversion to the UTC done? | Yes, I assume April 7$^{th}$ is meant. The Dobson measurements start at 7:29 CET at a mue-value of 3.24, which is sufficient for AD observations. |
| You should also add the uncertainty of each observation to the plots to show how different products compare. | Currently the absolute uncertainty of Brewers and Dobsons are not known and can therefore not be marked with error bars. The agreement of the Brewers and Dobsons are within 1% compared to the reference. We have added a citation regarding Brewers (Redondas et al. 2019) |
| Line 290 or part of Figure caption: "a worse performance" – why was the Dobson instrument's worse performance? | Rephrased |
| Line 293 "trends"-> results | Accepted |
| Line 294 " the least square fit is within 1 % over the whole measurement campaign" – Are you saying that every spectral fit was within 1 % of the observed spectrum or you are saying that the retrieval method that uses the LSF derived the TOC that was within 1 % of the Brewer-derived TOC? | We are saying that the fit of the plot stays within +/- 1% over the whole measurement campaign. We tried to make it clearer that this refers to the figure. |
| Figure 5 – why is the range of the individual differences (black squares) between Dobson and BTS is small in comparison to the Brewer/BTS comparisons (large spread in blue and green squares)? | This can be explained by the fact that BTS and Brewer deliver much more data points on each day, even for not ideal weather conditions (clouds, higher SZA, etc.) |
| This brings the question about the results shown in Figure 4. Does the histogram include the seasonal bias? | It includes all effects, so yes. |
| I wonder if you remove the seasonal bias (correct Dobson for the effective temperature bias) and repeat the histogram would the Gaussian shape be as wide? | We can assume that correction would improve the results. For this study we did not correct neither the Dobson or Brewer for stratospheric temperature. Redondas et al. 2014 addressed this question and Gröbner et al. 2021 recently presented Brewer and Dobson data including the stratospheric temperature effect correction (we added a citation). |
| Line 325 "percentual" -> percent? | Accepted |
| "overestimation of Koherent of a mean" ->" overestimation by Koherent on average by 1.64%" | Accepted |
| Line 327 "in the order as for" -> "comparable to" | Accepted |
| Figure 9: Histogram shows two distributions and incorporates the seasonal offset.  It is better to show comparisons for each season separately, similarly to what you are doing in Figure 10. | Yes, that would be an appropriate solution too. However, we wanted to show how it performs over the seasons without any stratospheric temperature correction |
| Line 323 "evidenced their performance" -> demonstrated instrument performance | Line 352: Accepted |
| Line 364 "simple modeling" – it would be useful to test the sensitivity of both TOC retrieval algorithms to the ozone profile shape. Most of the TOC retrievals (except in Antarctica during the spring ozone depletion) are not sensitive to the vertical ozone distributions except at large SZAs. | This could be done in further research. We thank for this suggestion, but this analysis exceeds the scope of this paper. |
| Line 370 change the to The at the beginning of the sentence | Accepted |

| | |
|---|---|
| Line 377 "relevant atmospheric parameters" – explain what you mean. Are you saying that the retrieval will be improved if aerosols and SO2 information would be available to constrain the spectral fitting? | The inclusion of measured aerosols or SO2 as input parameters was not investigated. Aerosols and Rayleigh are free fitting parameters of the least square fit. We have rephrased this part. |
| Line 378 – "actual atmosphere" -> observed atmosphere | See above. We have rephrased this part. |
| Line 387 "higher latitude"? | It should be higher SZA as in the original version of the manuscript. |
| Line 392 define "slightly" | This word is removed in the revised manuscript |
| Line 402 "linear trend"-> slope | Accepted |
| Line 404 "too high" – please define | See definition int the original manuscript in the brackets and this further explanation. Long term experience revealed that the single Brewer TOC drops already at an average AMF > 3.5 due to stray light effects, whereas a double Brewer with better stray light suppression is able to measure reliable TOC up to AMF = 4. We added a sentence to connect to this information. |
| Line 411 what do you mean by "calibration difficulty"? Please rephrase. | Accepted and rephrased |
| Line 425 and therefore comparable to Dobson? | Since we did not compare Koherent with Dobsons in Davos we cannot reliably cover such a statement. |
| I did not find information on where the data from these observational campaigns are archived or how these data can be obtained. | The data is available from: https://doi.org/10.6084/m9.figshare.14686656

We have added this information in the revised manuscript. |